# Subtyping Lung Cancer Using DNA Methylation in Liquid Biopsies

**DOI:** 10.3390/jcm8091500

**Published:** 2019-09-19

**Authors:** Sandra P. Nunes, Francisca Diniz, Catarina Moreira-Barbosa, Vera Constâncio, Ana Victor Silva, Júlio Oliveira, Marta Soares, Sofia Paulino, Ana Luísa Cunha, Jéssica Rodrigues, Luís Antunes, Rui Henrique, Carmen Jerónimo

**Affiliations:** 1Cancer Biology & Epigenetics Group-Research Center, Portuguese Oncology Institute of Porto (CI-IPOP), 4200-072 Porto, Portugal; sandra22nunes@gmail.com (S.P.N.); franciscadiniz93@gmail.com (F.D.); catarina.moreira.barbosa@gmail.com (C.M.-B.); veraconstancio24@gmail.com (V.C.); sofia.paulino@ipoporto.min-saude.pt (S.P.); analuisa.cunha@ipoporto.min-saude.pt (A.L.C.); 2Lung Cancer Clinic and Department of Medical Oncology, Portuguese Oncology Institute of Porto, 4200-072 Porto, Portugal; ana.v.silva@ipoporto.min-saude.pt (A.V.S.); julio.oliveira@ipoporto.min-saude.pt (J.O.); martasoares@ipoporto.min-saude.pt (M.S.); 3Department of Pathology, Portuguese Oncology Institute of Porto, 4200-072 Porto, Portugal; 4Department of Epidemiology, Portuguese Oncology Institute of Porto, 4200-072 Porto, Portugal; jessica.rocha.rodrigues@ipoporto.min-saude.pt (J.R.); luis.antunes@ipoporto.min-saude.pt (L.A.); 5Department of Pathology and Molecular Immunology, Institute of Biomedical Sciences Abel Salazar, University of Porto (ICBAS-UP), 4050-313 Porto, Portugal

**Keywords:** DNA Methylation, Lung Cancer, Subtyping, Cell-free DNA, Liquid Biopsy, Epigenetic Biomarker

## Abstract

Background: Lung cancer (LCa) is the most frequently diagnosed and lethal cancer worldwide. Histopathological subtyping, which has important therapeutic and prognostic implications, requires material collection through invasive procedures, which might be insufficient to enable definitive diagnosis. Aberrant DNA methylation is an early event in carcinogenesis, detectable in circulating cell-free DNA (ccfDNA). Herein, we aimed to assess methylation of selected genes in ccfDNA from LCa patients and determine its accuracy for tumor subtyping. Methods: Methylation levels of *APC*, *HOXA9*, *RARβ2,* and *RASSF1A* were assessed in three independent study groups (study group #1: 152 tissue samples; study group #2: 129 plasma samples; study group #3: 28 benign lesions of lung) using quantitative methylation-specific PCR. Associations between gene promoter methylation levels and LCa subtypes were evaluated using non-parametric tests. Receiver operating characteristic (ROC) curve analysis was performed. Results: In study group #2, *HOXA9* and *RASSF1A* displayed higher methylation levels in small-cell lung cancer (SCLC) than in non-small-cell lung cancer (NSCLC). *HOXA9* displayed high sensitivity (63.8%), whereas *RASSF1A* disclosed high specificity (96.2%) for SCLC detection in ccfDNA. Furthermore, *HOXA9* methylation levels showed to be higher in squamous cell carcinoma in comparison with adenocarcinoma in study group #1. Conclusions: Methylation level assessments in ccfDNA may provide a minimally invasive procedure for LCa subtyping, complementing standard diagnostic procedures.

## 1. Introduction

Lung cancer (LCa) is estimated to be the most commonly diagnosed cancer worldwide and the leading cause of cancer-related death in 2018 [1]. Most LCa cases are diagnosed at an advanced stage, endowing a modest five-year survival of 16%, despite continuing improvements in diagnosis and treatment [2,3]. Smoking is the most well-established LCa risk factor, as 85% of LCa cases are attributable to cigarette carcinogens, including benzopyrenes [4]. Interestingly, a shift in LCa topography and dominant subtype has been observed over the last 50 years due to alterations in cigarette manufacturing [5]. Indeed, decreased nicotine leads to increased puff volume, which translates into higher proportions of peripheral versus centrally located tumors [6]. LCa is generally divided into two major subtypes: small-cell lung cancer (SCLC, about 15% of all cases) and non-SCLC (NSCLC), which mostly comprises adenocarcinoma (AdC), squamous cell carcinoma (SCC), and large-cell carcinoma (LCC), as well as other less frequent histotypes [5]. Presently, AdC accounts for more than 40% of all LCa and has emerged as the main subtype because of the increasing access of tobacco smoke to the peripheral lung structures [5]. Although prognosis is strongly associated with stage, never-smokers and female patients usually endure better prognosis [7]. SCC constitutes, nowadays, the second most frequent LCa subtype, whereas SCLC is considered the most aggressive form of LCa (two-year survival rate of 10%), and both are also associated with tobacco smoke [5].

LCa subtyping is essential in treatment decisions and prognosis [8]. Indeed, the introduction of targeted therapy for epidermal growth factor receptor (*EGFR*), anaplastic lymphoma kinase (*ALK*), and ROS proto-oncogene 1 (*ROS1*) mutations increased the importance of discriminating AdC from SCC, since those mutations are highly associated with the former [9]. Additionally, with the recent introduction of the immune checkpoint inhibitors in NSCLC treatment, LCa subtyping became a crucial process for LCa treatment [9]. LCa diagnosis is usually based on pathological assessment of tissue fragments or cells collected either by bronchoscopy, fine-needle aspiration, or core-needle biopsy, depending on the tumor’s location and accessibility [10,11]. Nonetheless, in a sizeable proportion of cases, the material obtained is not sufficient for LCa subtyping [10], and the distinction between AdC and SCC may not be possible by morphology alone, even if assisted by immunohistochemistry [8]. Furthermore, these are invasive procedures amenable to complications such as hemorrhage and pneumothorax [8]. Molecular testing may provide higher specificity and decrease biopsy-associated risks. Indeed, an assay comprising the expression of eight microRNAS among four LCa subtypes (nonsquamous NSCLC, SCC, carcinoid, and SCLC) has shown high sensitivity and specificity, but it required tissue obtained from resections or biopsies and cytological samples, which might be difficult to obtain based on tumor location [12]. Thus, assays based on minimally invasive procedures are needed. Indeed, a study using a three-microRNA panel discriminated SCLC from NSCLC with high diagnostic accuracy in plasma samples [3].

LCa develops through a multistep process that includes altered DNA status [13]. Promoter hypermethylation of cancer-related genes is a common alteration, often associated with inactivation of tumor-suppressive genes [13], and its assessment has been proposed as a candidate biomarker for cancer detection and monitoring because of its stability and easy detection in tissue and body fluids [14,15]. Furthermore, the detection of circulating cell-free DNA (ccfDNA) methylation in plasma/serum samples may better represent tumor heterogeneity than tissue biopsy, being also less invasive [16]. Thus, we sought to evaluate the feasibility of using methylation of four gene promoters, previously characterized as hypermethylated in cancer [15,16,17,18,19], to discriminate among the major LCa subtypes in ccfDNA extracted from plasma, by means of quantitative methylation-specific PCR (qMSP). Selection of candidate genes *APC*, *HOXA9*, *RARβ2,* and *RASSF1A* was based on published literature [17,20,21,22], including our previous experience [15], since methylation levels disclosed differences among LCa subtypes, suggesting a role as biomarkers.

## 2. Materials and Methods

### 2.1. Patients and Sample Collection

Three independent study groups of LCa patients were included in this study. Study group #1 comprised 152 LCa patients diagnosed and treated at the Portuguese Oncology Institute of Porto (IPO Porto) between 2001 and 2016, from whom tissues were obtained from lung resection or tissue biopsy specimens. Study group #2 included 129 LCa patients primarily diagnosed at IPO Porto, between 2015 and 2017, from whom blood samples were collected before any treatment. Study group #3 comprised 28 plasma samples from patients suspect of harboring LCa but were found to carry benign lung disease, collected at IPO Porto between 2015 and 2019. Plasma was isolated from blood by centrifugation at 2000 rpm for 10 min at 4 °C, and subsequently frozen at −80 °C until further use. Tissue samples were obtained for each patient in study group #1, routinely fixed, and paraffin-embedded for standard pathological examination by H&E and specific immunostaining for tumor classification, grading, and staging [5,23,24]. Relevant clinical data were collected from clinical charts, and a database was constructed for statistical analysis purposes. This study was approved by the institutional review board of IPO Porto (Comissão de Ética para a Saúde, CES 120/2015). All patients enrolled signed informed consent according to the Declaration of Helsinki ethical principles.

### 2.2. DNA Extraction

Regarding study group #1, tumor areas were identified by an experienced pathologist in H&E slides. Eight micrometer tissue sections were cut, and tumor areas were macrodissected to maximize the proportion of malignant cells (> 70%), deparaffinized and rehydrated using xylene and 100% ethanol, respectively, and digested with 60 µL of proteinase K (20 mg/mL, Sigma-Aldrich^®^, Schnelldorf, Germany). DNA was extracted using a standard phenol-chloroform protocol previously described [25]. Concerning study groups #2 and #3, ccfDNA was extracted from 2 mL of plasma using QIAmp MinElute ccfDNA (Qiagen, Hilden, Germany), according to the manufacturer’s protocol and as previously described [15]. All extracted DNA was stored at −20 °C until further use.

### 2.3. Sodium Bisulfite Modification, Whole Genome Amplification (WGA), and DNA Quantification

Sodium bisulfite modification was performed using EZ DNA Methylation-Gold^TM^ (Zymo Research, Orange, CA, USA) according to the manufacturer’s recommendations. In tissue samples from study group #1, sodium bisulfite converted DNA was eluted in 60 µL of distilled water, whereas ccfDNA from study groups #2 and #3 were eluted in 10 µL of distilled water. One microgram of CpGenome^TM^ Universal Methylated DNA (Millipore, Temecula, CA, USA) was sodium bisulfite converted and used for control purposes. All sodium bisulfite converted DNA was stored at −80 °C until further use. WGA of ccfDNA extracted from study group #2 and #3 plasma samples was carried out using an EpiTect Whole Bisulfitome Kit (Qiagen, Hilden, Germany) according to the manufacturer’s instructions and as previously described [15]. Tissue-extracted DNA was quantified by a NanoDrop Lite Spectrophotometer (NanoDrop Technologies, Wilmington, DE, USA). Extracted, sodium bisulfite converted, and amplified ccfDNA was quantified using a Qubit 2 Fluorometer (Invitrogen, Carlsbad, CA, USA) following the manufacturer’s recommendations. The median ccfDNA concentration after extraction was 2.07 ng/µL (range: 0.392–26.6 ng/µL) and after WGA was 70.6 ng/µL (range: 0.756 to >120 ng/µL) for study group #2. Moreover, for study group #3, the median ccfDNA concentration after extraction was 0.660 ng/µL (range: 0.132–3.36 ng/µL) and after WGA was 60.3 ng/µL (range: 0.336–120 ng/µL).

### 2.4. Quantitative Methylation-Specific PCR (qMSP)

QMSP was performed to assess *APC*, *HOXA9*, *RARβ2,* and *RASSF1A* promoter methylation levels, and *β-Actin* served as a reference gene. For study group #1, sodium bisulfite modified DNA was used as a template, and reactions were carried out in 384-well plates using a LightCycler 480 Instrument (Roche Diagnostics, Manheim, Germany). Briefly, per well, 2 µL of modified DNA and 5 µL of KAPA SYBR® FAST qPCR Master Mix (Kapa Biosystems, MA, USA) were used. The primers’ volumes and conditions used for each gene are listed in Appendix A. Because of ccfDNA quantity limitations, in study groups #2 and #3, *APC*, *RASSF1A,* and *β-Actin* were run in a multiplex qMSP reaction using TaqMan probes and Xpert Fast Probe (GRISP, Porto, Portugal), whereas *RARβ2* and *HOXA9* were run in a separate qMSP reaction. Primers, probes, and fluorochromes used are listed in Appendix A. Six microliters of amplified ccfDNA was used as template, and the multiplex qMSP assays were carried out using 96-well plates in a 7500 Sequence Detector (Applied Biosystems, Perkin Elmer, CA, USA). All samples were run in triplicate, and a maximum 0.38 deviation between replicates was used. Amplification cycles above 40 were considered a “no result”, and these samples were not considered for further analysis.

Sodium bisulfite modified CpGenome^TM^ Universal Methylated DNA was subjected to a series of dilutions (5× dilution factor) and used to generate a standard curve allowing DNA relative quantification and plate efficiency assessment. Efficiency values above 90% were considered in each plate. An efficiency difference of maximum 5% between plates was considered. Relative methylation levels were generated calculating the ratio between the methylation levels of each gene and the respective value of *β-Actin* and multiplied by 1000 for easier tabulation [26,27].

### 2.5. Statistical Analysis

Non-parametric tests were used for comparing the distribution of methylation levels among different LCa subtypes and to evaluate associations with clinicopathological parameters (primarily by Kruskal–Wallis tests for three or more groups, followed by pairwise comparisons using Mann–Whitney U tests and Bonferroni’s correction, when applicable). The distribution of methylation levels was presented graphically in a scatter plot with a logarithmic scale. Zero values were transformed in value “1” for better representation and easier analysis [15]. A Spearman non-parametric test was performed to assess correlations between age and gene promoter methylation levels. Receiver operator characteristic (ROC) curve analysis was performed for each gene to assess biomarker performance. Samples were categorized as methylated (positive) or unmethylated (negative) based on the cutoff determined through ROC curve analysis, (i.e., the one providing the highest sensitivity and specificity, Youden’s J index) [28]. Validity estimates (sensitivity, specificity, and accuracy) were also determined. For this, multiple analyses of ROC curves generated via resampling randomly and dividing the sample into training (70%) and validation (30%) sets was performed. The cutoff value was estimated in the training set, and the validity estimates were calculated in the validation set using that cutoff. This procedure was repeated 1000 times, and the mean of the sensitivities and specificities was calculated, as previously described [15,29]. Gene panels were defined as positive when at least one of the genes was categorized as methylated (positive). The multiple ROC curve analysis was performed using R v3.4.4. Two-tailed *p* value calculations and ROC analysis were performed using a computer assisted program (SPSS Version 24.0, Chicago, IL, USA). Graphics were assembled with GraphPad 6 Prism (GraphPad Software, La Jolla, CA, USA).

## 3. Results

### 3.1. Clinical and Pathological Data

Relevant clinical and pathological data of study groups #1 and #2 are depicted in Table 1, whereas study group #3 information is displayed in Table 2. No correlations were found between patients’ age and gene promoter methylation levels in all study groups. Additionally, no differences were observed in the distribution of gender between study groups #2 and #3 (*p* = 0.391).

Associations between clinical stage and gene methylation levels were found, namely in study group #1 between stages I, II, and III; stage IV for *APC*, *RARβ2,* and *RASSF1A* (Appendix A); and for *RASSF1A* between stages III and IV in study group #2 (Appendix A).

### 3.2. Distribution of Gene Promoter Methylation Levels in ccfDNA

*APC, HOXA9, RARβ2,* and *RASSF1A* promoter methylation levels were evaluated in plasma samples from benign lung disease and LCa patients to assess whether they might differ among malignant and benign lung disease (Figure 1). *APC* and *RASSF1A* displayed higher methylation levels in LCa compared to benign cases (*p* = 0.006 and *p* = 0.033, respectively) (Figure 1), whereas *HOXA9* and *RARβ2* did not disclose significant differences (*p* = 0.329 and *p* = 0.133, respectively) (Figure 1).

Because LCa plasma samples displayed higher *APC* and *RASSF1A* methylation levels than benign lung disease cases in ccfDNA, their performance for LCa detection was assessed individually and in panel (Table 3, Appendix A). The panel comprising both genes disclosed 38.2% sensitivity and 92.8% specificity, corresponding to an overall accuracy of 47.6% for LCa detection (Table 3).

### 3.3. Gene Promoter Methylation Level Distribution Among Major LCa Subtypes 

In study group #1, SCLC disclosed significantly higher *APC*, *HOXA9*, *RARβ2,* and *RASSF1A* promoter methylation levels compared to NSCLC (*p* < 0.0001 for all genes, except for *HOXA9*, *p* = 0.021) (Figure 2A), whereas in ccfDNA (study group #2), only *HOXA9* and *RASSF1A* retained significant statistical differences (*p* < 0.0001 for both genes) (Figure 2B).

### 3.4. Biomarker Performance for SCLC Detection in Liquid Biopsies 

Since *HOXA9* and *RASSF1A* methylation levels were higher in SCLC in comparison with NSCLC in ccfDNA (study group #2), the performance of these genes for SCLC identification was evaluated (Table 4, Appendix A). *HOXA9* detected SCLC with 64% sensitivity, whereas *RASSF1A* individually disclosed 96% specificity (Table 4).

### 3.5. Gene Promoter Methylation Levels According to LCa Histological Subtypes

In study group #1, statistically significant differences among AdC, SCC, and SCLC were depicted for *APC*, *HOXA9*, *RARβ2,* and *RASSF1A* promoter methylation levels (*p* < 0.001 for all genes) (Figure 3A). In detail, *APC*, *RARβ2,* and *RASSF1A* showed statistically different methylation levels between AdC, SCC and SCLC (*p* < 0.0001) (Figure 3A), whereas *HOXA9* methylation levels were higher in SCC in comparison with AdC (*p* < 0.001) (Figure 3A). In ccfDNA (study group #2), promoter methylation levels of *HOXA9* and *RASSF1A* were higher in SCLC in comparison with AdC and SCC (*p* < 0.001) (Figure 3B). 

Because *HOXA9* methylation levels were significantly different between AdC and SCC in tissue samples, their biomarker performance for SCC detection in ccfDNA was evaluated. *HOXA9* hypermethylation detected SCC with 55.2% sensitivity, 74.3% specificity, and 71.6% accuracy, corresponding to an area under the curve (AUC) of 0.657 (Table 5Appendix A).

## 4. Discussion

Currently, LCa screening is only recommended for high-risk smokers; thus, most LCa cases are diagnosed at an advanced stage, entailing poor survival [2,30]. After LCa suspicion by imaging techniques, tissue biopsy and/or cytology are performed to confirm diagnosis and determine the LCa subtype [10]. Frequently, only a limited amount of material is obtained, which may result in the impossibility to perform a diagnosis or to adequately subtype the tumor, eventually leading to re-biopsy [31]. With the introduction of targeted therapies, a renewed interest in LCa subtyping emerged. In the presence of a positive biopsy, the LCa subtype can be determined by morphology in combination with immunohistochemistry [5]. Markers such as TTF1 for AdC and p40 for SCC are currently used for NSCLC subtyping [5,32]. Nevertheless, about 10% of NSCLC are rendered as “not otherwise specified” [5]. This may impact treatment decisions and patient prognoses, since AdC patients may benefit from targeted therapies for *EGFR* and *ROS1* or *ALK* mutations that significantly increase patient survival [33]. Additionally, therapy with pemetrexed or bevacizumab is contraindicated for SCC patients because of frequent complications [34,35]. Thus, correct LCa subtyping is critical to improve patient prognosis through a better selection of therapeutic strategies. Aberrant promoter methylation of cancer-related genes is a frequent and early alteration in carcinogenesis and may be easily detected in tissue and body fluids, constituting a potential biomarker for cancer detection and monitoring [15,26,36]. Hence, we aimed to determine whether promoter methylation of selected genes might allow for accurate discrimination among LCa subtypes in ccfDNA samples, providing a minimally invasive ancillary tool.

In tissue samples, higher methylation levels for all genes were found in SCLC compared to NSCLC, which is in line with previous reports on high *RASSF1A* and *RARβ2* methylation levels in SCLC cell lines and tissue samples [22,37,38,39,40]. Concerning *APC* promoter methylation, however, our results contradict previous studies that showed higher levels in NSCLC [39,41]. This discrepancy may be due to methodological differences, since conventional methylation-specific PCR (MSP), a qualitative analysis, was used in those studies, whereas we employed qMSP, a quantitative method with higher sensitivity and specificity [42]. High *HOXA9* methylation levels were previously reported in NSCLC [19,20,21,27], but to the best of our knowledge, this is the first study demonstrating higher *HOXA9* methylation levels in SCLC tissue and ccfDNA samples. Moreover, higher *HOXA9* methylation levels were found in SCC compared to AdC in tissue samples, confirming previous studies [43,44].

Concerning ccfDNA samples, *APC*, *RASSF1A,* and *HOXA9* were found hypermethylated in SCLC. In a previous study of ours, *APC* and *RARβ2* displayed higher methylation levels in SCLC compared to NSCLC in women [15]. The inclusion of both genders and a larger sample size of SCLC may explain these dissimilarities. Moreover, the higher *RASSF1A* methylation levels in ccfDNA samples from SCLC compared to NSCLC patients are in accordance with a previous study [45]. The differences in LCa stage distribution between study group #1 and study group #2 mainly are due to the nature of the samples, since study group #1 comprises tissue samples obtained mostly from surgical specimens, thus corresponding to earlier LCa stages. Furthermore, a small study group of patients with benign lung diseases was also included in this study. These patients had suspicious alterations detected in imaging exams that, consequently, were subjected to tissue biopsy, and LCa was not confirmed. Comparatively, LCa patients displayed higher methylation levels of *APC* and *RASSF1A* than those patients with benign lung disease, indicating that LCa detection may be achieved by analyzing DNA methylation in ccfDNA liquid biopsies, as previously demonstrated [15,17,27], complementing other diagnostic modalities (Figure 4). Nonetheless, these results need to be further validated in order to increase the assay sensitivity and evaluate their usefulness in clinical practice.

Several studies using microRNA (miRNA) panels attempted to discriminate among the different LCa subtypes in histological and cytological samples [12,46,47]. One of these panels, miRview^®^ lung (Rosetta Genomics), is based on the expression of eight miRNAs and displayed over 90% sensitivity for SCLC detection in tissue samples [12]. In our study, *HOXA9* displayed 64% sensitivity, whereas *RASSF1A* showed 96% specificity for SCLC detection in ccfDNA liquid biopsies. SCLC diagnosis using liquid biopsies in combination with clinical evaluation could allow for faster diagnosis and, consequently, sooner therapeutic decision, eventually precluding the need for an invasive procedure (i.e., biopsy).

Additionally, DNA methylation assessment in ccfDNA might aid in the diagnosis of tumors not assessable by the currently used techniques or in cases of limited diagnostic material, which do not allow for accurate diagnosis. To the best of our knowledge, this is the first study reporting a test assessing *HOXA9* and *RASSF1A* methylation levels for SCLC detection in liquid biopsies. In combination with clinical evaluation, including imaging and smoking status, this test may assist in LCa diagnosis (Figure 4). Thus, ccfDNA testing might be used as first line assay in the investigation of LCa suspects in combination with other diagnostic approaches, and a tissue biopsy would be performed only when it was negative for SCLC. In this scenario, NSCLC is likely to be present, and, as above mentioned, the differential diagnosis between AdC and SCC is not always possible (Figure 4). Interestingly, higher *HOXA9* methylation levels were depicted in SCC compared to AdC, although with lower sensitivity and specificity compared to miRview^®^ lung [12]. However, since we only analyzed one gene promoter, we are tempted to speculate whether our assay would be a faster and less expensive method for discriminating AdC from SCC. Nevertheless, further studies are required to assess its usefulness in a “real-world” scenario.

The main limitations of our study include the relatively small number of benign lung diseases and SCLC samples as well as the exclusion of other tumors with neuroendocrine differentiation such as carcinoids and large-cell neuroendocrine carcinoma. Nonetheless, this is an innovative proof-of-concept study that proposes LCa subtyping in a single blood analysis alongside clinical evaluation, complementing invasive tests to assist in patient diagnosis.

## 5. Conclusions

We report a methylation-based test for LCa subtyping in liquid biopsies. Such tests might not only speed up the diagnostic process, but they could also aid in clinical decision making. The clinical usefulness of these preliminary results requires further validation in larger independent cohorts.

## Figures and Tables

**Figure 1 jcm-08-01500-f001:**
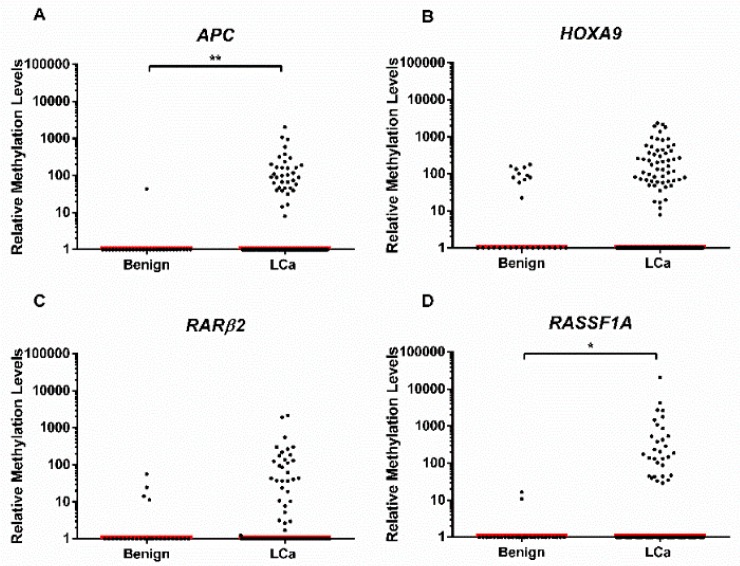
Scatter plot of (**A**) *APC* (number of values that fall in the *x* axis: benign = 27, LCa = 93), (**B**) *HOXA9* (number of values that fall in the *x* axis: benign = 17, LCa = 71), (**C**) *RARβ2* (number of values that fall in the *x* axis benign = 24, LCa = 92), and (**D**) *RASSF1A* (number of values that fall in the *x* axis: benign = 26, LCa = 98); methylation level distribution among lung cancer (LCa) patients from study group #2 (plasma samples) and benign lung diseases patients from study group #3 (plasma samples). Mann–Whitney U test. **p* < 0.05, ***p* < 0.01. Red horizontal line represents the median methylation level.

**Figure 2 jcm-08-01500-f002:**
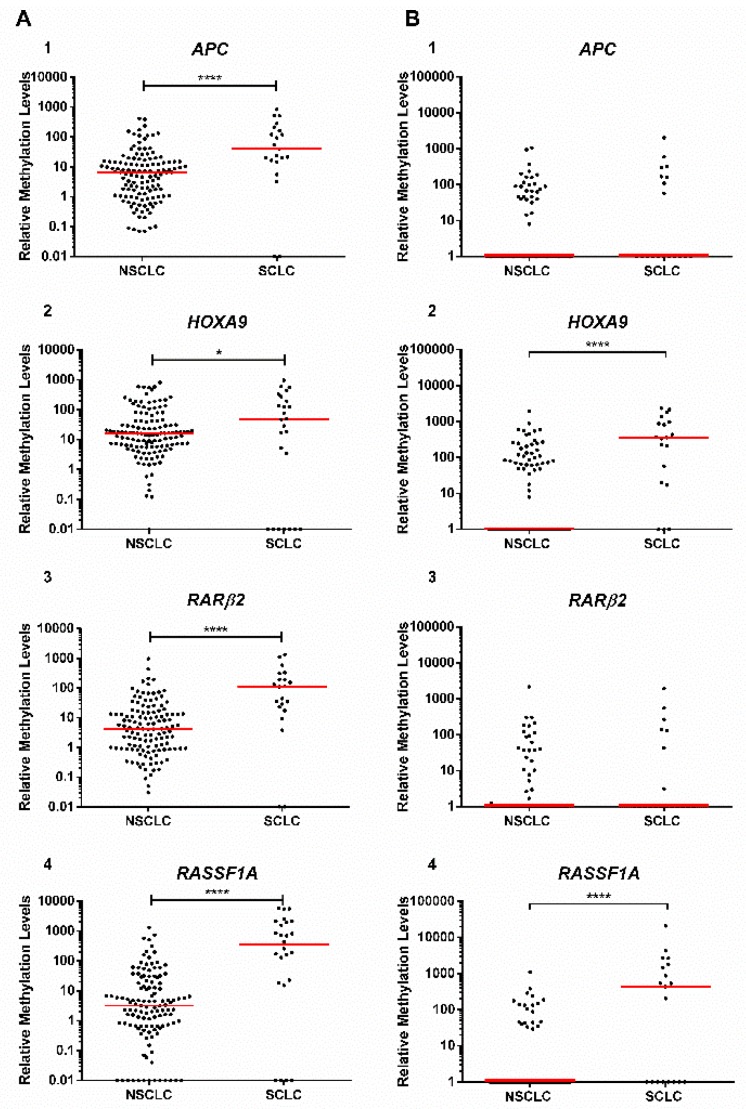
Scatter plot of (**A**) (1) *APC*, (2) *HOXA9*, (3) *RARβ2,* and (4) *RASSF1A* methylation level distributions among major lung cancer (LCa) subtypes (non-small-cell lung cancer (NSCLC) and small-cell lung cancer (SCLC)) in study group #1 (tissue samples) and (**B**) (1) *APC* (number of values that fall in the *x* axis: NSCLC = 82, SCLC = 11), (2) *HOXA9* (number of values that fall in the *x* axis: NSCLC = 68, SCLC = 3), (3) *RARβ2* (number of values that fall in the *x* axis: NSCLC = 84, SCLC = 12), and (4) *RASSF1A* (number of values that fall in the *x* axis: NSCLC = 90, SCLC = 8) among NSCLC and SCLC in study group #2 (plasma samples). Mann–Whitney U test. **p* < 0.05 and *****p* < 0.0001. Red horizontal line represents the median methylation level.

**Figure 3 jcm-08-01500-f003:**
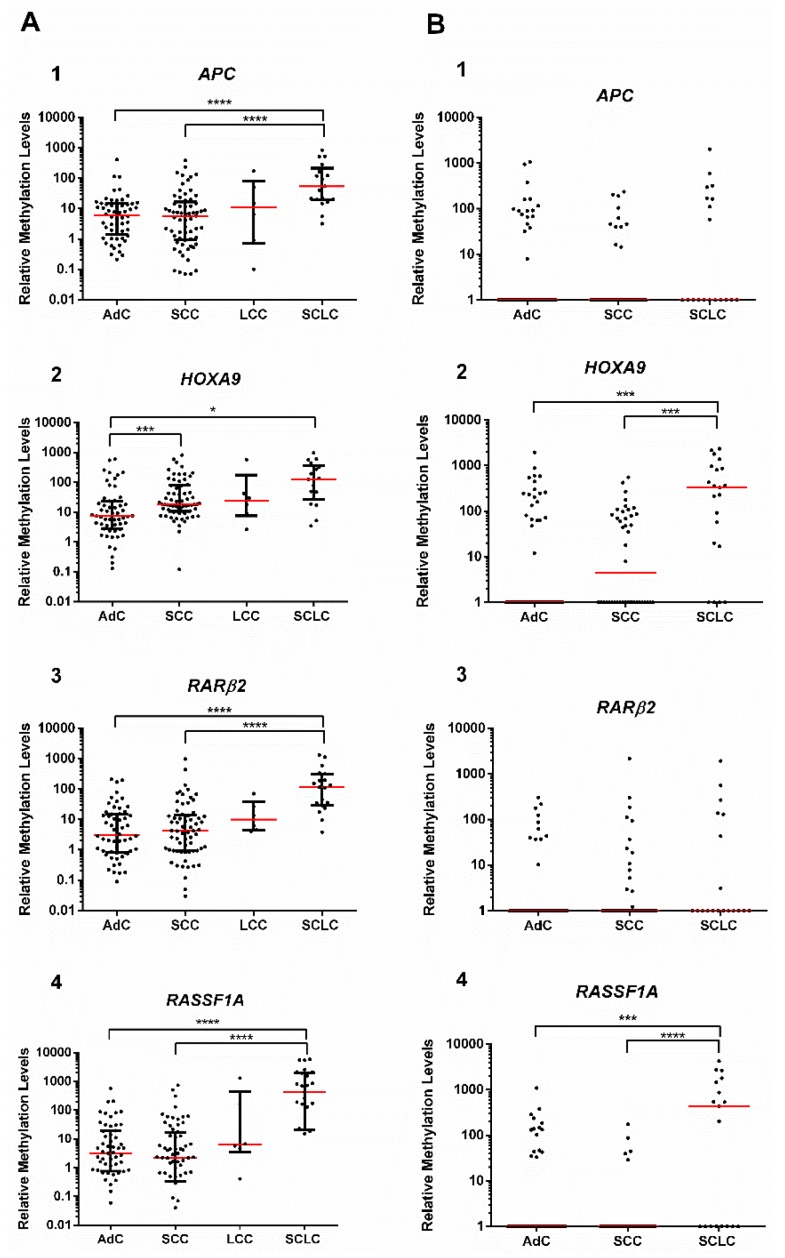
Scatter plot of (**A**) (1) *APC*, (2) *HOXA9*, (3) *RARβ2,* and (4) *RASSF1A* methylation level distributions among lung cancer (LCa) histological subtypes (adenocarcinoma (AdC), squamous cell carcinoma (SCC), large cell carcinoma (LCC), and small-cell lung cancer (SCLC)) in study group #1 (tissue samples) and (**B**) (1) *APC* (number of values that fall in the *x* axis: AdC = 48, SCC =31, SCLC = 11), (2) *HOXA9* (number of values that fall in the *x* axis: AdC = 44, SCC = 21, SCLC = 3), (3) *RARβ2* (number of values that fall in the *x* axis: AdC = 54, SCC = 27, SCLC = 12), and (4) *RASSF1A* (number of values that fall in the *x* axis: AdC = 50, SCC =37, SCLC = 8) methylation level distributions among LCa histological subtypes (AdC, SCC, and SCLC) in study group #2 (plasma samples). Kruskal–Wallis test, followed by pairwise comparison though Mann–Whitney U and Bonferroni’s correction. **p* < 0.05, ***p* < 0.01, ****p* < 0.001, and *****p* < 0.0001. Red horizontal line represents the median methylation level.

**Figure 4 jcm-08-01500-f004:**
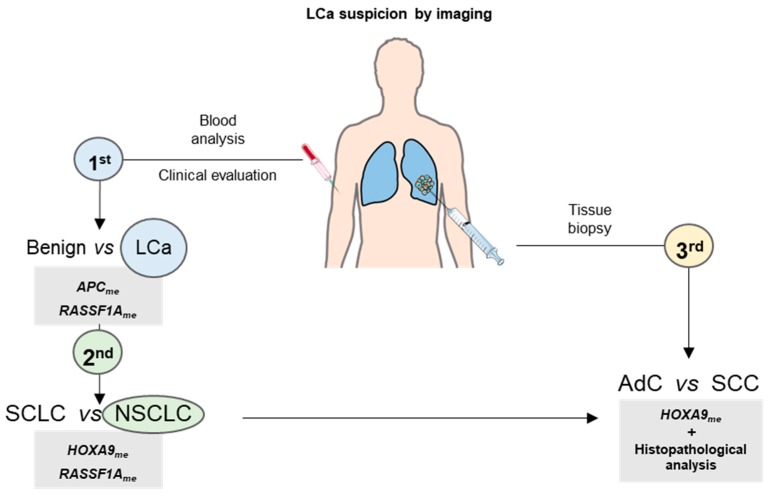
Proposed algorithm for lung cancer (LCa) subtyping when LCa is suspected by imaging. Firstly, blood analysis by assessing *APC* and *RASSF1A* methylation levels could be performed in combination with clinical evaluation in order to detect LCa. Then, the methylation levels of *HOXA9* or *RASSF1A* could be evaluated to aid in determining the major LCa subtype present (non-small-cell lung cancer (NSCLC) vs small-cell lung cancer (SCLC)). After, if NSCLC present, a tissue biopsy could be performed, and *HOXA9* methylation levels could aid histopathological analysis in discriminating between adenocarcinoma (AdC) and squamous cell carcinoma (SCC).

**Table 1 jcm-08-01500-t001:** Clinicopathological features of lung cancer (LCa) patients from study groups #1 and #2.

Clinicopathological Features	Study Group #1	Study Group #2
	Tissue Samples	Plasma Samples
**Patients (n)**	152	129
**Gender**		
Male (M)	113	91
Female (F)	39	38
**Age, median (range)**	65 (45–83)	66 (38–89)
**Histological Subtype, n; %^1^**		
AdC	56 (26 M; 30 F); 39%	65 (35 M; 30 F); 50%
SCC	65 (62 M; 3 F); 41%	42 (39 M; 3 F); 31%
LCC	6 (6 M); 4%	-
SCLC	25 (19 M; 6 F); 16%	19 (16 M; 3 F); 15%
**Clinical stage**		
I	74	15
II	33	11
III	24	27
IV	21	76

^1^ Includes 3 non-small-cell lung carcinoma (NSCLC) in study group #2. Abbreviations: AdC, adenocarcinoma; F, female; LCC, large-cell carcinoma; m, Male; SCLC, small-cell lung cancer; SCC, squamous cell carcinoma; n.a., not available.

**Table 2 jcm-08-01500-t002:** Clinicopathological features of benign lung disease patients from study group #3.

Clinicopathological Features	Study Group #3
	Benign Lung Diseases Plasma Samples
**Patients (n)**	28
**Gender**	
Male (M)	22
Female (F)	6
**Age, median (range)**	63 (40–86)
**Benign lung disease**	
Inflammatory processes	9
Tuberculosis	3
Chondroid hamartoma	4
Silicosis	1
Fibrosis	1
Without evidence of disease	10

**Table 3 jcm-08-01500-t003:** Biomarker performance of promoter gene methylation for LCa detection in plasma samples.

Genes	AUC ^1^	Sensitivity %	Specificity %	Accuracy %
*APC*	0.622	25.0	96.4	37.3
*RASSF1A*	0.591	23.7	95.4	36.0
*APC/RASSF1A*	-	38.2	92.8	47.6

^1^ AUC—area under the curve.

**Table 4 jcm-08-01500-t004:** Biomarker performance of promoter gene methylation for SCLC detection in study group #2 (plasma samples).

Genes	AUC ^1^	Sensitivity %	Specificity %	Accuracy %
*HOXA9*	0.805	63.9	84.2	82.2
*RASSF1A*	0.747	52.0	96.2	79.1

^1^ AUC—area under the curve.

**Table 5 jcm-08-01500-t005:** Biomarker performance of *HOXA9* promoter gene methylation for SCC detection in study group #1 (tissue samples).

Validity Estimates	*HOXA9* Methylation
Sensitivity %	55.2
Specificity %	74.3
Accuracy %	71.6
AUC	0.657

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
