# Peer review of "Subtyping Lung Cancer Using DNA Methylation in Liquid Biopsies"

_jcm, 2019, doi:10.3390/jcm8091500_

Round 1
Reviewer 1 Report
General:
Nunes et al. analyzed the DNA methylation levels of the genes APC, HOXA9, RARβ2,and RASSF1in tissue specimens of lung cancer patients as well as circulating cell-free DNA from lung cancer patients and hospital controls. The aim was to evaluate the markers for minimally invasive subtyping of lung cancer.
The manuscript is well written but requires a number of improvements/corrections and would benefit from more details in some parts. In general, there is an urgent need for noninvasive or minimally invasive diagnostic tools for diagnosis of lung cancer. The presented biomarkers (well-known markers like RASSF1as well as the novel maker HOXA9) might serve as candidates for a future marker panel, however, the data do not fully support the claims made by the authors.
Major:
The performance of biomarkers (sensitivity & specificity etc. as defined by ROC analysis) in cancer diagnostics is critical for its possible application in clinical practice. Lung cancer is a deadly disease, frequently with dismal prognosis. Therefore, a high specificity of individual biomarkers is important in order to avoid false-positive results that could result in false treatments or cause psychological distress. A single biomarker is unlikely to achieve sufficient sensitivity, thus necessitating panels composed of several individual markers with high specificity. Depending on the target population (high-risk (like heavy smokers) or general population), the required specificities might range from 97% to more than 99%. Before subtyping can be achieved, lung cancer diagnosis should be established.
The authors provide selected sensitivities & specificities in Tables 3, 4, & 5, however, it would be much more informative to have complete ROC curves that allow to judge the marker performance over the full range of sensitivities & specificities, especially in the high specificity range.
– Please provide ROC curves for all markers and their combinations as depicted in Tables 3-5. The curves could be placed in the Supplementary Figures file.
It is somewhat frustrating to interpret Figures 1 & 2 (in part also Figure 3) because many data points (and the medians) lie on the x axis. The comparison of benign cases (just 28 cases) and lung cancer cases (129 cases) is particularly difficult because the “cohort” of benign cases is so much smaller.
– Please improve Figures 1 & 2 accordingly
– Please increase, if possible, the group of patients with benign diseases. A 1:1 matching regarding gender, age and smoking status would greatly improve the comparison with the lung cancer group. Because COPD is considered a “precursor” of lung cancer, patients with this disease should be included as well.
In my experience, the isolation and bisulfite conversion of ccfDNA only leaves minuscule amounts of DNA for downstream applications. The EpiTect kit requires at least 50 ng of bisulfite converted DNA. Also, WGA might introduce bias. The following determination of methylation levels can therefore lead to quite different results when repeating experiments (or lead to no results at all).
You used a Qubit fluorometer to quantify the DNA yield. Please provide median & range of the yields (or at least some typical yields) for the tumor group and the benign disease group before and after WGA (tumors usually have more ccfDNA).
You did triple measurements with your qMSP. How reproducible were your measurements? Please provide some details.
Instead of absolute methylation level you used relative levels. Please provide more details or cite papers that use this approach (if available). Why are so many data points exactly 1 in Figure 1 & 2? I would expect some more variation. Or does this reflect the fact that many measurements did not yield a result because of limited amounts or quality of DNA (see comments above). It would be best to provide an Excel table with the raw and normalized data etc.
Figure 4 and the interpretation of the results in the Discussion part suggest that the markers can be used to distinguish (1) between benign and lung cancer, (2) between SCLC and NSCLC, as well as (3) between AdC and SCC.
(1) 38% sensitivity & 93% specificity are not sufficient. In addition, the control group (benign lung diseases) was too small (see comments above).
(2) HOXA9reaches 64% sensitivity & 84% specificity and RASSF152% sensitivity & 96% specificity, which has some potential but is not sufficient individually. On the other hand, the combination HOXA9/RASSF1does not show a better performance – so why is it listed in Figure 4?
(3) I cannot really see (from the data in Figure 3 and Table 5) why HOXA9can distinguish AdC from SCC (there is too much overlap). “Statistically different” does not necessarily mean that individual cases can be distinguished in clinical practice. Besides, the title of the manuscript states subtyping in liquid biopsies. Here, you switch back to subtyping in tissues.
Minor:
You use the term “cohort” for your patient groups. In view of the epidemiological definition of this term (usually a much larger group of people who share certain characteristics) I would encourage to replace “cohort” with “patient group”, “study group” or just “group”.
Table 3: Why is the AUC value for the combination APC/RASSF1missing?
Table 4: Why is the AUC value for HOXA9/RASSF1missing?
In some figure legends and the main text (e.g., lines 197 - 199), the gene names should be in italics.
The official gene name is RASSF1(RASSF1Ais an older alias).
Lines 220-223: You use the term “discriminated”. This is an interpretation of the results. It would be more accurate to write “showed statistically different methylation levels between … and …”.
Line 278-279: Why correspond surgical specimens (group #1) to earlier LCa stages compared to blood samples (group #2) that could have been drawn before surgery - was the blood drawn post surgery?
Line 48: the comma before “in 2018” can be omitted
Line 70: …on the tumor’s …
Line 95: … diagnosed and treated …
Line 96: … from whom tissues …
Line 115: … to the manufacturer’s …
Line 246: … the LCa subtype …
Line 246: …allied? Do you mean “in combination with”?
Line 287: … miRNAs …
Line 292: … in the diagnosis …
Conclusions:
The authors should tone down their interpretation and conclusions regarding the usefulness of the described markers. After addressing the above points, the manuscript by Santos et al. can be recommended for publication in Journal of Clinical Medicine.
Author Response
General:
Nunes et al. analyzed the DNA methylation levels of the genes APC, HOXA9, RARβ2, and RASSF1 in tissue specimens of lung cancer patients as well as circulating cell-free DNA from lung cancer patients and hospital controls. The aim was to evaluate the markers for minimally invasive subtyping of lung cancer.
The manuscript is well written but requires a number of improvements/corrections and would benefit from more details in some parts. In general, there is an urgent need for noninvasive or minimally invasive diagnostic tools for diagnosis of lung cancer. The presented biomarkers (well-known markers like RASSF1as well as the novel maker HOXA9) might serve as candidates for a future marker panel, however, the data do not fully support the claims made by the authors.
Major:
The performance of biomarkers (sensitivity & specificity etc. as defined by ROC analysis) in cancer diagnostics is critical for its possible application in clinical practice. Lung cancer is a deadly disease, frequently with dismal prognosis. Therefore, a high specificity of individual biomarkers is important in order to avoid false-positive results that could result in false treatments or cause psychological distress. A single biomarker is unlikely to achieve sufficient sensitivity, thus necessitating panels composed of several individual markers with high specificity. Depending on the target population (high-risk (like heavy smokers) or general population), the required specificities might range from 97% to more than 99%. Before subtyping can be achieved, lung cancer diagnosis should be established.
The authors provide selected sensitivities & specificities in Tables 3, 4, & 5, however, it would be much more informative to have complete ROC curves that allow to judge the marker performance over the full range of sensitivities & specificities, especially in the high specificity range.
– Please provide ROC curves for all markers and their combinations as depicted in Tables 3-5. The curves could be placed in the Supplementary Figures file.
R: The authors thank the Reviewer for the comment and for the opportunity to improve our work. As stated in the paper (Page 4, lines 153-157), samples were defined as methylated (positive) or unmethylated (negative) based on the cut-off determined through ROC curve analysis, i.e., the point comprising the highest sensitivity and specificity. The cut-offs were defined for each gene individually and gene panels were defined as positive when, at least, one of the genes was considered methylated (positive). This information was added in the Experimental Section, Statistical analysis, Page 4, lines 160-161. Thus, ROC curves for the individual genes were added as Supplementary Material, Figures S1, S2 and S3.
It is somewhat frustrating to interpret Figures 1 & 2 (in part also Figure 3) because many data points (and the medians) lie on the x axis. The comparison of benign cases (just 28 cases) and lung cancer cases (129 cases) is particularly difficult because the “cohort” of benign cases is so much smaller.
– Please improve Figures 1 & 2 accordingly
R: As required by the reviewer Figures 1, 2 and 3 were improved by adding the number of cases that fall in the x axis in the figure’s caption.
– Please increase, if possible, the group of patients with benign diseases. A 1:1 matching regarding gender, age and smoking status would greatly improve the comparison with the lung cancer group. Because COPD is considered a “precursor” of lung cancer, patients with this disease should be included as well.
R: The authors thank the reviewer for the comment. Indeed, all the samples included were diagnosed as benign lung diseases by histopathological examination available in our Biobank. These patients correspond to suspicious alterations in imaging that, consequently, led to tissue biopsy and LCa was not confirmed. Although we were not able to retrieve cases of COPD at our Center (because it is a Cancer Institute), we plan to collect this type of samples or establish collaborations with other hospitals to test them it in the future.
Ideally, matching for gender, age and smoking status would be preferable. Although age differed between cancer and benign patients (p=0.024), no correlations were found with genes’ methylation levels, as stated in the manuscript page 4 lines 167-168 (see table below).
Tumor samples:
|
Correlation |
|||||||
|
|
Age |
APC |
RASSF1A |
HOXA9 |
RARB2 |
||
|
Spearman rho |
Age |
Correlation coefficient |
1,000 |
-,003 |
-,061 |
-,097 |
-,057 |
|
Sig. (2-tail) |
. |
,973 |
,495 |
,275 |
,524 |
||
|
N |
129 |
129 |
129 |
129 |
129 |
||
Benign samples:
|
Correlation |
|||||||
|
|
Age |
APC |
RASSF1A |
HOXA9 |
RARB2 |
||
|
Spearman rho |
Age |
Correlation coefficient |
1,000 |
,143 |
-,016 |
,229 |
,195 |
|
Sig. (2-tail) |
. |
,468 |
,935 |
,240 |
,320 |
||
|
N |
28 |
28 |
28 |
28 |
28 |
||
Furthermore, we did not observe differences in the distribution of gender and smoking status between cancer and benign patients (please see tables below).
|
Tumor_vs_Benign * Gender |
|||||
|
|
Gender |
Total |
|||
|
M |
F |
||||
|
Tumor_vs_Benign |
Benign |
n |
22 |
6 |
28 |
|
% Tumor_vs_Benign |
78.6% |
21.4% |
100.0% |
||
|
Tumor |
n |
91 |
38 |
129 |
|
|
% Tumor_vs_Benign |
70.5% |
29.5% |
100.0% |
||
|
Total |
n |
113 |
44 |
157 |
|
|
% Tumor_vs_Benign |
72.0% |
28.0% |
100.0% |
||
Qui-square p value: 0.391
|
Tumor_vs_Benign * Smoke |
|||||
|
|
Smoke |
Total |
|||
|
YES |
NO |
||||
|
Tumor_vs_Benign |
Benign |
n |
22 |
6 |
28 |
|
% Tumor_vs_Benign |
78.6% |
21.4% |
100,0% |
||
|
Tumor |
n |
100 |
29 |
129 |
|
|
% Tumor_vs_Benign |
77.5% |
22.5% |
100,0% |
||
|
Total |
n |
|
35 |
157 |
|
|
% Tumor_vs_Benign |
|
22.3% |
100.0% |
||
Qui-square p value: 0.904
In my experience, the isolation and bisulfite conversion of ccfDNA only leaves minuscule amounts of DNA for downstream applications. The EpiTect kit requires at least 50 ng of bisulfite converted DNA. Also, WGA might introduce bias. The following determination of methylation levels can therefore lead to quite different results when repeating experiments (or lead to no results at all).
R: The authors thank the reviewer for the comment. In this study we have used the EpiTect kit to perform whole genome amplification aiming to increase qMSP sensitivity since we extracted ccfDNA from low plasma quantities. We used a kit specifically for sodium-bilsulfite sequences with a multiple displacement amplification, which includes a DNA polymerase with exonuclease proofreading activity to decrease biases, and random primers that amplify both unmethylated and methylated sequences. Reports show that WGA yield unbiased amplified products amenable for methylation analysis when the DNA quantity is limited (Bundo 2012, Clinical Epigenetics; Mill 2006, Biotecniques). In the PCR reaction, 3 replicates were used per sample. Moreover, each sample was only considered for analysis when the amplification difference between replicates was up to a maximum of 0.38. Moreover, amplification cycles above 40 were considered as a “no result” and were not considered for further analysis.
You used a Qubit fluorometer to quantify the DNA yield. Please provide median & range of the yields (or at least some typical yields) for the tumor group and the benign disease group before and after WGA (tumors usually have more ccfDNA).
R: The authors thank the reviewer for the suggestion. Indeed, we observed higher DNA quantities in tumor samples (before WGA median: 2.07 ng/µL range: 0.392 – 26.6 2.07 ng/µL; after WGA median: 70.6 ng/µL range: 0.756->120 ng/µL) than in benign tumor samples (before WGA median: 0.660 ng/µL range: 0.132 – 3.36 ng/µL; after WGA median: 60.3 ng/µL range: 0.336-120 ng/µL).
You did triple measurements with your qMSP. How reproducible were your measurements? Please provide some details.
R: For each plate, WGA amplified CpGenome Universal Methylated DNA was subjected to a serial of dilutions (1:5) that were used to generate a standard curve. Only efficiencies above 90% for all genes were considered for further analysis. An efficiency difference of maximum 5% between plates and a 0.38 of deviation between the 3 replicates was used.
Instead of absolute methylation level you used relative levels. Please provide more details or cite papers that use this approach (if available).
R: The authors thank reviewer for the comment that allowed us for further elucidate on this matter. The relative methylation levels are based on the ratio between the mean methylation levels and the respective value for B-Actin (housekeeping gene). This is a widely used method (Hoque 2006, Journal of Clinical Oncology; Henrique 2007, Clinical Cancer Research; Hulbert 2016; Clinical Cancer Research, among many others).
Why are so many data points exactly 1 in Figure 1 & 2? I would expect some more variation. Or does this reflect the fact that many measurements did not yield a result because of limited amounts or quality of DNA (see comments above). It would be best to provide an Excel table with the raw and normalized data etc.
R: The authors thank the reviewer for the comment. However, we presented a scatter plot with a log-10 scale for samples’ relative methylation levels graphical representation. The data points represented as 1 correspond to samples that had B-Actin (the housekeeping gene), but did not display any relative methylation levels for the analyzed genes (APC, RASSF1A, RARβ2 and HOXA9), being classified as “negative” or “0”. For a better representation of the relative methylation levels distribution, we have used a log-10 scale and represent these samples as 1, so they can be shown on the graph. Samples in which B-Actin did not amplify were not included in the analysis.
Figure 4 and the interpretation of the results in the Discussion part suggest that the markers can be used to distinguish (1) between benign and lung cancer, (2) between SCLC and NSCLC, as well as (3) between AdC and SCC.
(1) 38% sensitivity & 93% specificity are not sufficient. In addition, the control group (benign lung diseases) was too small (see comments above).
R: The authors thank the reviewer for the comment. Our major aim was to subtype lung cancer using DNA methylation in liquid biopsies. We have included a small group of patients with benign lung disease as proof-of-concept to demonstrate that in our clinical setting methylation levels analysis could be an additional tool to detect malignancy in patients with suspicious imaging. Indeed, our group has already tackled this issue in a previously published study (Nunes 2018, Cancers).
(2) HOXA9 reaches 64% sensitivity & 84% specificity and RASSF1 52% sensitivity & 96% specificity, which has some potential but is not sufficient individually. On the other hand, the combination HOXA9/RASSF1 does not show a better performance – so why is it listed in Figure 4?
R: In fact, the combination does not improve the performance of each gene individually. Hence the panel was deleted. However, we would like to emphasize that, individually, HOXA9 methylation levels display high sensitivity, considering that is just one gene analyzed in plasma samples (64%), whereas RASSF1A methylation levels disclosed high specificity (96%). Moreover, ROC curve analysis demonstrated statistical significance (HOXA9, p<0.0001; RASSF1A p=0.0006).
(3) I cannot really see (from the data in Figure 3 and Table 5) why HOXA9 can distinguish AdC from SCC (there is too much overlap). “Statistically different” does not necessarily mean that individual cases can be distinguished in clinical practice. Besides, the title of the manuscript states subtyping in liquid biopsies. Here, you switch back to subtyping in tissues.
R: The authors respectfully disagree with the reviewer. Squamous cell carcinoma samples showed statistically significant higher methylation levels than adenocarcinoma (p=0.0008). Although, graphically, the difference between the two groups in these figures may not be very obvious, the medians are statistically different (according to non-parametric Mann-Whitney U test, which is very stringent). Furthermore, the statistical analysis performed was again thoroughly revised and validated by an expert in biomarker performance statistics before manuscript re-submission. Additionally, ROC curve analysis is statistically significant (AUC=0.657, p=0.001).
Our main results show that we were able to distinguish SCLC from NSCLC in liquid biopsies. Nevertheless, when a NSCLC is present, we suggest that a tissue biopsy could be performed. In this scenario, since differential diagnosis between adenocarcinoma and squamous cell carcinoma is not always possible, methylation levels analysis in tissue could aid histopathological subtyping, as stated in the manuscript page 10, Figure 4.
Minor:
You use the term “cohort” for your patient groups. In view of the epidemiological definition of this term (usually a much larger group of people who share certain characteristics) I would encourage to replace “cohort” with “patient group”, “study group” or just “group”.
R: The authors thank the reviewer for the comment and the term “cohort” was changed to “study group”.
Table 3: Why is the AUC value for the combination APC/RASSF1missing?
R: As stated before, samples were defined as methylated (positive) or unmethylated (negative) based on the cut-off determined through ROC curve analysis. The cut-offs were defined for each gene individually and gene panels were defined as positive when at least one of the genes was considered methylated (positive). Because logistic regression was not used for these calculations, the AUC of the panel cannot be provided.
Table 4: Why is the AUC value for HOXA9/RASSF1missing?
R: The panel HOXA9/RASSF1A was removed from the manuscript following Reviewer # 1 suggestion.
In some figure legends and the main text (e.g., lines 197 - 199), the gene names should be in italics.
R: We thank the reviewer for the opportunity to improve our manuscript. The names were changed to italic.
The official gene name is RASSF1 (RASSF1Ais an older alias).
R: The authors thank the reviewer for the comment. In this study, we analyzed RASSF1 isoform A that differs from other isoforms (Malpeli 2019, Cancers). Below you can find a schematic representation of the RASSF1 locus from Malpeli 2019, Cancers.
Lines 220-223: You use the term “discriminated”. This is an interpretation of the results. It would be more accurate to write “showed statistically different methylation levels between … and …”.
R: The comment was changed accordingly.
Line 278-279: Why correspond surgical specimens (group #1) to earlier LCa stages compared to blood samples (group #2) that could have been drawn before surgery - was the blood drawn post surgery?
R: The authors thank the reviewer for the comment. Blood samples from group #2 were collected before surgery or any treatment. This study group corresponds to a consecutive series of blood samples independent from study group #1. Since blood samples were collected at the time of diagnosis, the majority of patients were diagnosed with advanced stage disease, as expected. In contrast, the majority of tissue samples of study group #1 comprises early stages (non-invasive and non-metastatic tumors) since they correspond to specimens obtained by surgery from patients with localized disease. This is discussed in page 9, lines 284-287.
Line 48: the comma before “in 2018” can be omitted; Line 70: …on the tumor’s …; Line 95: … diagnosed and treated …; Line 96: … from whom tissues …; Line 115: … to the manufacturer’s …; Line 246: … the LCa subtype …; Line 246: …allied? Do you mean “in combination with”?; Line 287: … miRNAs …; Line 292: … in the diagnosis …
R: The authors thank the reviewer for the opportunity to improve the manuscript. The required changes were included in the manuscript in respective lines.
Conclusions:
The authors should tone down their interpretation and conclusions regarding the usefulness of the described markers. After addressing the above points, the manuscript by Nunes et al. can be recommended for publication in Journal of Clinical Medicine.
R: The authors thank the reviewer for the opportunity to improve the manuscript. The required changes were included in the manuscript in respective lines.
Reviewer 2 Report
In the manuscript entitled "Subtyping lung cancer using DNA methylation in liquid biopsies" by Nunes et al., the authors describe results of methylation sensitive RT-PCR analyses of 4 genes for lung cancer subtyping. The authors analysed 3 different cohorts: FFPE lung cancer specimens, blood samples from lung cancer patients and blood samples from patients with benign lung disease. Methylation of HOXA9 and RASSF1A were found to differ between SCLC and NSCLC samples. Further, methylation of HOXA9 distinguished adenocarcinoma and squamous cell carcinoma samples. Overall, the manuscript is well written but the presented data are not really new.
Comments:
1. Did the authors design the RT-PCR assays by themselves or are the assays already published? Which strategy was used to validate the accuracy of the assays?
2. Based on which criteria were the 4 genes selected? There are numerous datasets published where methylation was compared between either SCLC and NSCLC or between NSCLC subtypes. Methylation of APC, RASSF1A and RARß was very frequently analysed in lung cancer patients in the past (both in primary tumor tissue and in serum), thus, the manuscript lacks novelty.
3. Stage of disease distribution between cohort #1 and cohort #2 is very different. #1 contains mainly early stage patients, #2 contains mainly late stage patients. It is known that methylation of some genes is a very early event in lung cancer pathogenesis and methylation changes of other genes is a late event. Thus, cohorts should be as comparable as possible.
4. I suggest to combine Figure 2 and supplementary Figure 1.
5. Figure 3: Y-axes and plot types of panels A and B differ.
6. Correlation of methylation data with clinico-pathological data from the patients is missing.
7. Is there HOXA9 methylation difference also present in TCGA´s LUAD and LUSC data?
Reviewer 3 Report
The authors aimed to assess methylation of selected genes in circulating cell-free DNA from lung cancer patients and determine its accuracy for tumour subtyping. They studied three independent cohorts using quantitative methylation-specific PCR and showed that methylation level assessment may provide a minimally-invasive procedure for lung cancer detection and subtyping.
In literature, different molecular biomarkers have been tested as alternative or complementary diagnostic tools to obtain a higher sensitivity in the early diagnosis of lung cancer. Among the wide range of molecular markers,
epigenetic markers are most frequently
investigated and seem to be most promising because of their crucial role in the cell cycle. In particular, both hypomethylation and hypermethylation of well-known cancer-related genes have been found to be a process occurring in the early stages of cancer development [Balgkouranidou I. et al. Lung cancer epigenetics: Emerging biomarkers. Biomark. Med. 2013, 7, 49–58; Li C.M. et al. Current and future molecular diagnostics in non-small-cell lung cancer. Expert Rev. Mol. Diagn. 2015, 15, 1061–1074; Calabrese F. et al., Are There New Biomarkers in Tissue and Liquid
Biopsies for the Early Detection of Non-Small Cell
Lung Cancer? J. Clin. Med. 2019, 8, 414]. However, to date some pitfalls concerning the design and overall validation of biomarkers have certainly delayed their transfer to the clinical setting.
The present study can be considered interesting in the field because it includes both lung cancer patients (tissue or plasma samples) and patients with benign lung diseases (plasma samples).
MAJOR CONCERNS
1) The article is generally well written, I would suggest to tone down the importance of DNA methylation analyses in the diagnostic flow chart. Indeed, on the basis of these data, I would suggest to give importance to the role of methylation status in early lung cancer diagnosis. For what concerns subtyping, morphological (histological and immunohistochemical) approaches represent the gold standard and molecular investigations should be performed always in association to them, not alone (see 2nd step in Figure 4 and discussion).
2) Do you have any experience in "paired" gene methylation evaluation (both in circulating cell-free DNA and in tumour tissue samples taken from the same patient)?
MINOR CONCERNS
1) Title of Table 2: remove "LCa"
2) Some grammar and typo errors are present
Author Response
Reviewer #3
The authors aimed to assess methylation of selected genes in circulating cell-free DNA from lung cancer patients and determine its accuracy for tumour subtyping. They studied three independent cohorts using quantitative methylation-specific PCR and showed that methylation level assessment may provide a minimally-invasive procedure for lung cancer detection and subtyping.
In literature, different molecular biomarkers have been tested as alternative or complementary diagnostic tools to obtain a higher sensitivity in the early diagnosis of lung cancer. Among the wide range of molecular markers, epigenetic markers are most frequently investigated and seem to be most promising because of their crucial role in the cell cycle. In particular, both hypomethylation and hypermethylation of well-known cancer-related genes have been found to be a process occurring in the early stages of cancer development [Balgkouranidou I. et al. Lung cancer epigenetics: Emerging biomarkers. Biomark. Med. 2013, 7, 49–58; Li C.M. et al. Current and future molecular diagnostics in non-small-cell lung cancer. Expert Rev. Mol. Diagn. 2015, 15, 1061–1074; Calabrese F. et al., Are There New Biomarkers in Tissue and Liquid Biopsies for the Early Detection of Non-Small Cell Lung Cancer? J. Clin. Med. 2019, 8, 414]. However, to date some pitfalls concerning the design and overall validation of biomarkers have certainly delayed their transfer to the clinical setting.
The present study can be considered interesting in the field because it includes both lung cancer patients (tissue or plasma samples) and patients with benign lung diseases (plasma samples).
MAJOR CONCERNS
1) The article is generally well written, I would suggest to tone down the importance of DNA methylation analyses in the diagnostic flow chart. Indeed, on the basis of these data, I would suggest to give importance to the role of methylation status in early lung cancer diagnosis. For what concerns subtyping, morphological (histological and immunohistochemical) approaches represent the gold standard and molecular investigations should be performed always in association to them, not alone (see 2nd step in Figure 4 and discussion).
R: The authors thank the reviewer’s positive evaluation and the suggestions. Herein, our major aim was to be able to subtype lung cancer using DNA methylation in liquid biopsies. We have included a small group of benign lung diseases patients as proof-of-concept to show that in our clinical setting methylation levels analysis could be an additional tool to detect malignancy in patients with suspicious imaging. Indeed, our group has already tackled this issue in a previously published study (Nunes 2018, Cancers).
Histopathological subtyping has important implications in therapeutic decisions, however it requires material collection through invasive procedures. Moreover, often only a minute amount of material is obtained and in about 10% of cases, NSCLC are classified “not otherwise specified”. Thus, non-invasive tools that may aid in lung cancer subtyping, allied with other approaches including histopathological analysis, are of major relevance. Accordingly, Figure 4 was altered to clarify the benefit of complementing the standard diagnostic strategies with, possibly, DNA methylation analyses.
2) Do you have any experience in "paired" gene methylation evaluation (both in circulating cell-free DNA and in tumour tissue samples taken from the same patient)?
R: Although this study was performed in two different cohorts of patients, we are currently collecting liquid biopsies and tissue samples from the same patients, as it would be interesting to compare directly methylation levels in different types of samples from the same patient.
MINOR CONCERNS
1) Title of Table 2: remove "LCa"
R: We thank the reviewer for helping us to improve our manuscript. LCa was removed.
2) Some grammar and typo errors are present
R: The manuscript was thoroughly revised and the errors/typos found were corrected.
Round 2
Reviewer 1 Report
General:
Nunes et al. have addressed most of the points made in my first review. They provided additional data, including previously missing ROC curves that allow a better evaluation of the results.
However, the authors have not sufficiently toned down the overly optimistic interpretation of their data.
Major:
Please rewrite the Results/Conclusions in the Abstract.
– The sentence “HOXA9andRASSF1A disclosed high sensitivity and specificity for SCLC detection in ccfDNA” is plainly wrong: According to the ROC curves, the markers might either reach high specificity and low sensitivity – or (relatively) high sensitivity and low specificity – but never high sensitivity andspecificity.
– The sentence “Furthermore, HOXA9 methylation levels discriminated squamous cell carcinoma from adenocarcinoma…” is also incorrect. As I stated in my initial review, the data does not support this conclusion. Again, the usefulness of p-values is generally overrated and overinterpreted in biomarker research. Besides, an AUC of 0.657 is rather underwhelming. With that kind of performance, I cannot see any application in clinical practice.
The ROC curves in Suppl. Fig. S1 do not match the high specificities given in the corresponding Table 3.
– Which information is wrong? Please correct.
In your response you explained how you calculated the methylation levels. However, I cannot find lines 153-157 or 160-161 in the revised manuscript.
Also, lines 167-168 do not correlate with your statement (perhaps, the line numbers shifted during conversion of the Word file to PDF?). Furthermore, I cannot find lines 284-287 and other locations stated.
In your response, you provide more experimental details, e.g. regarding WGA and yields, but I am missing this information in the manuscript.
– Please include the experimental details in the manuscript, especially the yields and the reproducibility of repeat measurements. Low yields from plasma can result in inconclusive or missing methylation data, which in turn might limit the usefulness of the markers.
You explained, in your response, why so many data points are exactly 1. That information is also missing in the manuscript.
– Please add this information in the experimental part of the manuscript or at least provide it as supplemental information (this should include additional citations)
Is it possible that you accidentally uploaded an older version of the corrected manuscript because so much of the new information is missing in the manuscript?
Conclusions:
Whereas the authors have improved the manuscript, they still have not heeded my main point, to tone down their interpretation / conclusions regarding the usefulness of the described markers in the abstract. This should definitely be addressed.
Author Response
General:
Nunes et al. have addressed most of the points made in my first review. They provided additional data, including previously missing ROC curves that allow a better evaluation of the results.
However, the authors have not sufficiently toned down the overly optimistic interpretation of their data.
R: The authors thank the reviewer for the comment. Several alterations in results interpretation were performed in order to tone down the conclusions (See Discussion Section and Figure 4). Nonetheless, as stated in the Discussion (Page 10, Lines 326-328-Track Changes- simple markup”) “This is an innovative proof-of-concept study that proposes LCa subtyping in a single blood analysis, complementing invasive tests to assist in patient diagnosis.”
Major:
Please rewrite the Results/Conclusions in the Abstract.
– The sentence “HOXA9 and RASSF1A disclosed high sensitivity and specificity for SCLC detection in ccfDNA” is plainly wrong: According to the ROC curves, the markers might either reach high specificity and low sensitivity – or (relatively) high sensitivity and low specificity – but never high sensitivity and specificity.
R: The authors thank the reviewer for the comment and for the opportunity to improve our work. The Results section of the abstract was reformulated to “HOXA9 displayed high sensitivity (63.8%) whereas RASSF1A disclosed high specificity (96.2%) for SCLC detection in ccfDNA” Page 9, Lines 306-307-Track Changes- simple markup”).
– The sentence “Furthermore, HOXA9 methylation levels discriminated squamous cell carcinoma from adenocarcinoma…” is also incorrect. As I stated in my initial review, the data does not support this conclusion. Again, the usefulness of p-values is generally overrated and overinterpreted in biomarker research. Besides, an AUC of 0.657 is rather underwhelming. With that kind of performance, I cannot see any application in clinical practice.
R: The authors thank the reviewer for the comment and for the opportunity to clarify this issue. In the Abstract section, the statement was changed to “Furthermore, HOXA9 methylation levels showed to be higher in squamous cell carcinoma in comparison with adenocarcinoma in study group #1”. HOXA9 methylation levels showed to be a promising biomarker to aid in lung cancer subtyping, however we agree that it is not ready to use in clinical practice, since additional studies are required to evaluate its usefulness in a “real-world” scenario. A statement was added in the Discussion section (Page 10, lines 322-323-“Track Changes- simple markup”).
The ROC curves in Suppl. Fig. S1 do not match the high specificities given in the corresponding Table 3.
– Which information is wrong? Please correct.
R: The authors thank the reviewer for the comment and for the opportunity to improve our work. The ROC curves were corrected accordingly (new Supplementary Figure 1).
In your response you explained how you calculated the methylation levels. However, I cannot find lines 153-157 or 160-161 in the revised manuscript.
Also, lines 167-168 do not correlate with your statement (perhaps, the line numbers shifted during conversion of the Word file to PDF?). Furthermore, I cannot find lines 284-287 and other locations stated.
R: We indicated the lines according to the Word file- track changes “simple markup”, but indeed, in the PDF file downloaded the alterations did not correspond to those lines because ii was “all markup”. We apologize for that. Nevertheless, the alterations were made already in the previous revised version and to avoid any confusion they can now be found in the lines according to the command “Track Changes- simple markup marked in yellow (current version: Page 4, lines 152-154 or 159-161)
In your response, you provide more experimental details, e.g. regarding WGA and yields, but I am missing this information in the manuscript.
– Please include the experimental details in the manuscript, especially the yields and the reproducibility of repeat measurements. Low yields from plasma can result in inconclusive or missing methylation data, which in turn might limit the usefulness of the markers.
R: The authors thank the reviewer for the comment and for the opportunity to clarify this issue. We added additional information in the Experimental Section, namely the DNA concentrations and further details in data and statistical analysis (Page 3, lines 129-133, Page 4, lines 146-148, 152-154 and 159-161 – “Track Changes- simple markup”).
You explained, in your response, why so many data points are exactly 1. That information is also missing in the manuscript.
– Please add this information in the experimental part of the manuscript or at least provide it as supplemental information (this should include additional citations)
R: The authors thank the reviewer for the comment. This information was added to the Experimental Section, Page 4, lines 159-161 and 168-170- “Track Changes- simple markup”).
Is it possible that you accidentally uploaded an older version of the corrected manuscript because so much of the new information is missing in the manuscript?
R: The authors thank the reviewer for the comment. We did not upload an older version of the manuscript. The difference is due to the formatting of the track changes (“all markup” vs “simple markup”). We apologize for the confusion, but we did not realize it. Nevertheless, we added additional information to the Experimental Section and altered some of our conclusions in the Discussion section, as recommended. You can find the information with Word command “Track Changes- simple markup” marked in yellow.
Conclusions:
Whereas the authors have improved the manuscript, they still have not heeded my main point, to tone down their interpretation / conclusions regarding the usefulness of the described markers in the abstract. This should definitely be addressed.
R: The authors thank the reviewer for the comment. As stated above, we have toned down the interpretation and conclusions (please see comments above).
Reviewer 2 Report
The authors addressed my concerns.
Author Response
Dear reviewer,
Thank you for your comments which allow us to improve the manuscript.